# Low-Digit and High-Digit Polymers in the Origin of Life

**DOI:** 10.3390/life9010017

**Published:** 2019-02-02

**Authors:** Peter Strazewski

**Affiliations:** Institut de Chimie et Biochimie Moléculaires et Supramoléculaires (Unité Mixte de Recherche 5246), Université de Lyon, Claude Bernard Lyon 1, 43 bvd du 11 Novembre 1918, 69622 Villeurbanne CEDEX, France; strazewski@univ-lyon1.fr; Tel.: +33-472-448-234

**Keywords:** digit multiplicity, information, transmission, encoding, translation, diversity, function

## Abstract

Extant life uses two kinds of linear biopolymers that mutually control their own production, as well as the cellular metabolism and the production and homeostatic maintenance of other biopolymers. Nucleic acids are linear polymers composed of a relatively low structural variety of monomeric residues, and thus a low diversity per accessed volume. Proteins are more compact linear polymers that dispose of a huge compositional diversity even at the monomeric level, and thus bear a much higher catalytic potential. The fine-grained diversity of proteins makes an unambiguous information transfer from protein templates too error-prone, so they need to be resynthesized in every generation. But proteins can catalyse both their own reproduction as well as the efficient and faithful replication of nucleic acids, which resolves in a most straightforward way an issue termed “Eigen’s paradox”. Here the importance of the existence of both kinds of linear biopolymers is discussed in the context of the emergence of cellular life, be it for the historic orgin of life on Earth, on some other habitable planet, or in the test tube. An immediate consequence of this analysis is the necessity for translation to appear early during the evolution of life.

## 1. Introduction

The analogy between the role of linear biological polymers in cellular life, and that of strings composed of digits in the elaboration and transmission of discrete information, is as old as the foundations of information theory [1,2,3] and the Central Dogma of molecular biology [4,5,6]. The definition and understanding of “information” in the former theory finds its analogy in the latter through a shift on a superordinate level, in a metaphor [7]. Classical information theory presupposes that the transmission of information is based on a purely one-way “Laplacian” deterministic transfer mechanism whereby the instruction encoded in a one-dimensional (“linear”) digital information determines fully the outcome; it implies causality and, in principle, “bottom-up” predictability. This of course is not the case in living organisms. Even in the most simple organism there are, despite the Central Dogma stating a strict irreversibility of information transmission, feedback mechanisms that “impose” changes in the “programme” of the descendants of the very organism, through the selection of stochastic errors, and thus permit its “evolution” in a Darwinian sense, hence, a change in frequency and abundance of a heritable trait of a population, through adaptation and in competition (“fitness”) with others, as opposed to through random drift, migration or molecular changes per se. Therefore, the metaphorical use of information theoretical or mathematical terms like “programme”, “code”, “signal”, “noise”, “random”, “algorithm” for the description of physical, chemical and biological phenomena and processes need to be taken with uttermost care and a full awareness of the pros and cons of metaphors across these research domains.

Having said that, the advent of an upcoming new non-deterministic information theory, owing to expected developments in the artificial intelligence (AI) field, will probably induce a general overhaul of classical information theoretical terms, since life-like extrinsic feedback mechanisms are not only inevitably emerging but also at the very heart of AI [8]. Likewise, in a number of expected processes, that are thought to have taken place shortly before and during the (or any other) orgin of life, the differences between genotype and phenotype are expected to be less pronounced and the strength of intrinsic and extrinsic feedback mechanisms weaker than they are in extant biota [9,10]. Therefore, it is worthwhile to keep analysing the effects of linear biological polymers in the light of information theory [11]. Here the consequences of different multiplicities of digits in linear polymers (strings) are examined with respect to the information transmission from one to another biopolymer and their respective functional competences. It will become apparent that both low-digit and high-digit linear polymers are likely to be prerequisite for any life form to emerge from complex prebiotic chemical systems.

## 2. Information Transmission and Capacity of Digital Strings

According to Claude Shannon [3], the information content of a string of *M* digits to be transmitted, termed here *Shannon Information (SI)*, is inversely related to its expectancy of appearing by chance from a random alignment of its digital components. *SI* is thus a measure of “unpredictability” and “randomness”, since the more unlikely a particular sequence of *M* digits is to self-assemble by pure chance the more its occurrence is “informative” [7] and “patterened” [11]. The amount of this information stored in a string of binary digits is proportional to the logarithm of *N* possible states of that system, denoted *log*_2_
*N* (Equation (1)). Changing the base of the logarithm to a different number *b* has the effect of multiplying the value of the logarithm by a fixed constant *log*_2_
*b*. The choice of the base *b* determines the unit used to measure information, for which different unit names are used: *bit/shannon* (binary digit) for *b* = 2, *nit/nat/nepit* (natural or Neperian digit) for *b* = e ≈ 2.718, *trit* (trinary or ternary digit) for *b* = 3, *quit* (quaternary digit) for *b* = 4, *dit/ban/hartley* (decimal digit) for *b* = 10, and so forth. One *nit* ≈ 1.443 *bits*, 1 *trit* ≈ 1.585 *bits*, 1 *quit* = 2 *bits*, 1 *dit* ≈ 3.322 *bits*, etc. *M* denotes the number of digits in a string (Equation (2)). The longer the string the proportionally higher the *SI* (Equation (3)).
*SI(N)* = *log*_2_*N* = *log_b_ N* · *log*_2_*b*(1)
*N* = *b^M^*(2)
*SI(M)* = *M* · *log*_2_*b*(3)

The digit multiplicity *b*, that is, the number of different digits in a string, can in principle vary without limitation. The higher the multiplicity *b* the shorter the string with identical information storage capacity and *SI* (Table 1). For example, 40-meric *bit* strings (*SI* = 40 *bits*) can realise about a (long-scale) trillion different variants. Roughly the same information storage capacity is realised by their “compression” to 26-meric *trit* strings, 20-meric *quit* strings, 18-meric *quint* strings, and so forth. The higher multiplicities chosen in Table 1 (*b* > 5) refer to the possible generation of “secondary” high-digit strings from the “primary” low-digit strings through the usage of a code that enhances the multiplicities *b* to higher *b’* values by integer exponents {2, 3, 4}: 2^3^ = 8, 3^2^ = 9, 2^4^ = 4^2^ = 16, 3^3^ = 27, 4^3^ = 64, 3^4^ = 81, 4^4^ = 256 (see also Figure 1). Another multiplicity denoted *b’_eff_* refers to a partly redundant use of a higher multiplicity *b’, vide infra*.

The usage of information theoretical *SI* for the calculation of the “unpredictability” in biopolymers of extant biota, such as polynucleotides and polypeptides, is in that sense questionable as all evolved life forms did not emerge from random self-assemblies of its monomeric residues; on the contrary, they evolved from deletions, insertions and rearrangments of whole DNA segments from “horizontal gene transfer”, and from “random walks through sequence-space” being carried over in minute mutation steps from antecedent organisms to their progeny that happened to be “fit” enough to give viable offspring again. However, in the context of an origin of first cellular life from complex chemical systems, “random” self-assemblies—with all constraints imposed by the real atomistic chemistry that differentiates this kind of randomness from true randomness in an abstract mathematical sense—could be reasonably plausible events happening in a prebiotic environment, given recurrent chemical potential gradients and the prebiotic availability of sufficiently large amounts of linear polymers of some realistic distribution of limited lengths *M*. Therefore, it makes sense to analyse processes that could have occurred on the early Earth, or that might occur under other comparable circumstances, in the light of information transmission according to the formalism pioneered by Shannon, as well as purely combinatorial information storage capacity assuming equal probabilities for all string sequences.

## 3. Low-Digit Memory Polymers

The bricks that biotic nature—as we know it on Earth—uses to maintain a systemic memory throughout many generations of reproduction of individual system units, that themselves individually almost fully degrade and ultimately vanish, are composed of nucleic acids. Nucleic acids can harbor and transmit an astonishingly large number of information through more or less faithfully copying long strings termed “linear polymers”. In biotic nature these strings are very soluble and solvent-accessible polyanionic linear polymers composed of 4 different (but similar) “letters”. In information theoretical terms these are *quit*s (not Qbits) realised by the nucleotides A, G, C and U or T. Natural *quits* are pairwise complementary to one another through the Watson–Crick rules (G–C, A–U or A–T), which gives the grounds for faithful template-directed copying as during cellular replication (double-copying of complementary single-stranded DNA), or complement-copying, as for the transcription or reverse-transcription of strings of nucleic acids of virtually deliberate length (from DNA to RNA or vice versa). Upon translation, in contrast, specific “coding fractions” of these strings of *quits*, rather than being recognized one by one as during complement-copying, can be read out by anticodons—parts of transfer RNA bound to ribosomes—as a series of consecutive *3-letter* “words” termed “base triplets”, that is, information theoretical unitary blocks of 3-*quit* “quytes” (3*Q*), that are chained up in heterogeneously and almost deliberately long “sentences” termed “reading frames” (genes). The grammar, syntax and dialects (gene regulation, message editing, epigenetics) used in these sentences are then a matter of *system unit type* and *network organization*, for instance, cell (germ line or somatic), organism, species, interaction with other species, ecological traits and niches, and so forth.

Biotic nucleic acids such as RNA and DNA, whether translated or not, are used in known animate systems as *quit* carriers that are relatively easy to copy through molecular templating, irrespective of whether this copying is assisted by enzymes or not. The information theoretical difference between enzyme-catalyzed or enzyme-free (“spontaneous”) template copying is merely the fidelity resulting in a more or less complete carry over of the information from template polymer to product polymer. Complementary or self-complementary read-outs, that is, the copying and encoding rules as we know them from biotic genome replication, transcription and the “universal genetic code”, are reducible to the hydrogen bond donor-acceptor patterns that are being exposed from the so-called Watson–Crick face of the natural nucleobases of each nucleotide [12,13]. These patterns are not limited to the natural nucleobases. Other N-heterocycles may furnish different patterns and thus distinct pairing preferences [14]. Therefore, from a purely chemical point of view, molecular template variants like binary- or ternary-digit memory polymers composed of strings of subsequent *bits* or, respectively, *trits* are well imaginable (Figure 1). These *bits* or *trits* could be complementary through different pairing modes. In principle, each *bit* or *trit* could be strictly self-complementary, bearing an exclusively self-recognising pairing property: 0 pairs only with 0, 1 pairs only with 1, 2 pairs only with 2. Chemically much more likely, alternative *bit* genomes could be composed of only, for instance, G and C or only A and U, where one digit (0 or 1) is complementary to the other (Figure 1A). Alternative *trit* genomes could bear two digits that are complementary to one another (e.g. 0 pairs with 2) and a third strictly self-complementary digit (1 pairs only with 1), thus being composed of, say, G, C and X, the latter being an exclusively self-recognising nucleotide (Figure 1B).

In addition, the coding fractions of such low-digit memory polymers could be read out, for example, as 4-*bit bytes* (4*B*) or 3-*trit trytes* (3*T*). Biotic 3-*quit quytes* (3*Q*: natural base triplets) comprise *b’* = *b*^3^ = 4^3^ = 64 different values, thus offer 64 different triplet “codons” (large frame in Figure 1C). So do 6-*bit bytes* of binary memory polymers (6*B*: *b’* = 2^6^ = 64, not shown) but such long codons would necessitate hexaplet anticodons for translational read-out. Shorter 5-*bit bytes* (5*B*) generate *b’* = 32 different pentaplet codons. Chemically more realistic are 4-*bit bytes* (4*B*) giving rise to *b’* = 16 different quadruplet codons and 3-*bit bytes* (3*B*) giving merely *b’* = 8 different triplet codons (Figure 1A). In ternary-digit memory polymers, blocks of 3-*trit trytes* (3*T*) produce *b’* = 3^3^ = 27 different triplet codons, whereas 4-*trit trytes* (4*T*) generate *b’* = 3^4^ = 81 different quadruplet codons. The latter set of codons would suffice for an even larger than natural (biotic) diversity of translated digits *b’*, compare the 4*T* code in Figure 1B with the 3*Q* code in Figure 1C. Of note, the information storage capacity is invariant irrespective of the type of code used to compact the low-digit into a high-digit string, cf. identical left and right values *b^M^* and *SI* before and, respectively, after translation, e.g. *b^M^* = 2^28^ = 4^14^ = 16^7^; 3^28^ = 9^14^ = 81^7^; 4^28^ = 16^14^ = 256^7^ (see Figure 1A–C for *SI* values).

These are simply numerical-combinatorial guidelines that exempt “degenerate” (redundant) and “stop” codons. The modern-day ribosomal translation mechanism has established a universal genetic code based on 64 different 3-*quit quytes*, i.e., triplet codons that are currently occupied by merely 20 “proteinogenic” amino acids and usually 3 stop codons, unless a biocompatible “expanded alphabet” for triplet codons has been artificially introduced at selected positions using synthetic nucleotides that offer a distinct “orthogonal” pairing selectivity that may differ from the natural Watson–Crick rules [12,15,16,17,18,19,20]. Most of the twenty proteinogenic amino acids are encoded by a set of faster and slower, thus, more or less erroneously translated, redundant codons being read by more abundant and, respectively, rarer “isoaccepting” anticodon triplets all carrying the same amino acid. Hence, the multiplicity *b’* in the secondary “condensed” high-digit polymer is reduced to an effective high-digit value *b’_eff_* = 20. Already the fact that the effectively used multiplicity in extant biota is less than a third of the theoretically possible (20 amino acids + 1 stop/64 codons) hints at a limit that organic molecules encounter. It is the recognition selectivity, the uniqueness and reliability of a specific molecular recognition that becomes increasingly ambiguous and error-prone with growing diversity of the digits [9,11]. This is the information theoretical ground for the “central dogma” of molecular biology to be a correct assumption [4,5,6]. Nature can reliably transmit information, being imprinted into molecular atom arrangements under liquid water-conditions only from low-digit to low-digit, or from low-digit to high-digit polymers, never from high-digit to low-digit polymers. These low-digit read-outs from high-digit polymers would immediately loose their informational identity. Molecular recognition of high-digit polymers, thus from highly diverse molecular variants, is too ambiguous.

## 4. High-Digit Functional Polymers

The copying and encoding principles shown in Figure 1 insinuate that unbranched molecular strings (1D polymers) composed of a limited number of different monomeric complementary residues (monomers), that is, strings that bear a relatively low multiplicity of digits (low *b*), are likely to be a general feature of memory keepers in any animate system. The lower the digit multiplicity the simpler the composition of the template and less ambiguous it is to copy and replicate the string on a molecular level [22]. This generates fewer errors in template-copied memory polymers, thus, a higher replication error threshold for a given spreading “quasi-species” (similar genome population), and eventually imposes a weaker selection pressure on the maximal genome length of any evolved organism [23,24,25].

The opposite is true when it comes to functional translation products, in which the higher their digit multiplicity is (high *b’*) the stronger the compression, the shorter the resulting string lengths (lower *M’*). A comparison between primary low-digit and secondary high-digit polymers of the same string length (when *M* = *M’*) reveals a much higher structural diversity of the latter. This high diversity is further multiplied by the number of reading frame shifts (*M_b_*) that give rise to an encoded set of a completely different choice of translated string sequences (Figure 1A–C, below each code). This generates translated string polymers that are inherently difficult to copy through direct templating, since the complement rules—analogous to the Watson–Crick base pairing rules—needed to be as manifold and exclusive as the digits are diverse. On the other hand, the longer the translation blocks (codons, translated words) in the messenger nucleic acids the more compact is the generated diversity of the secondary polymer, which allows for more diverse “molecular functions” at a given secondary string length *M’*. A higher compositional diversity means a wider, more versatile and fine-grained (higher dimensional) sequence space, thus lending such polymers easier access to their folding and assembly into structurally more defined, more rigid, catalytically more competent functional objects [9]. 

## 5. Discussion: What to Expect from Linear Biopolymers of Unknown Biota

At unchanged *SI* the compression *M_b = 2_*/*M_b ≥ 2_* of a string of digits upon enhancement of the digit multiplicity *b* follows a binary-logarithmic dependence (Equation (4), numerical examples in Table 1).
*M_b = 2_*/*M_b ≥ 2_ = log*_2_*b*(4)

All digital devices are based on *bit* string information storage and transmission systems. The lowest possible digit multiplicity works best despite the resulting longest possible string length *M*. Not only are uncompressed *bit* strings highly unpredictable in Shannon’s sense. Historically, electronic devices work most reliably when the digits are encoded by a “weak current” of whatever strength {1} and “no current” {0}. The storage and transmission of this kind of string is least error-prone, since the difference between “zero” and “more than zero” is the largest possible, so a binary digital read-out delivers the highest signal-to-noise ratio [11]. If the digits were “tension”, one could feed computers with *trit* string instructions based on sequences of “no tension” {0}, “positive tension” {1} and “negative tension” {2} of whatever strength. The resulting strings would be *log*_2_3-fold shorter, the information more compact, but also more error-prone to transmit. Dangerously unreliable would be the usage of *quit* strings in digital devices. The digits would have to be realised from a “highest current”, “high current”, “low current” and “no current” code. Replace “current” and “tension” with “amplitude” and, respectively, “frequency (wavelength)” or “phase transition” (liquid-solid, amorphous-crystalline, absorbing-reflective, and so forth), and the same applies to optical data storage devices. Human minds, at the other extreme, can easily distinguish all *dits* from one another (and more). It is all a matter of the distinguishability and of the similarity of the digits.

Biotic nature has hitherto evolved *quit* strings as macromolecular memory carriers, why *b* = 4? With respect to *bit* strings this means a two-fold compression. As long as we cannot precisely measure the similarity of molecular digits (but see [11]), a general quantified answer remains elusive. These primary *quit* strings code in parts for *vigintit* strings, *b’_eff_* = 20, which means that the translated parts are furthermore two-fold compressed isoinformational secondary polymers (four-fold with respect to *bit* strings). To obtain a quantitative answer, why *quit*-to-*vigintit* and not any other low digit-to-high digit translation, is impossible by analytical means owing to the complexity of extrinsic and intrinsic feedback networks, as mentioned in the introduction. However, as for the population dynamics of replicators [26], the dynamics of the stochastic generation of translation products is best approached by simulation methods from differential equations, particularly of the kind, where the translation fidelity comes out “impedance-matched” to that of the replication of the whole genome [9,10].

Generally, the reason why *quit* strings have evolved to reach a stable dynamic optimum in extant biota has a strong bearing with “Eigen’s paradox”, which states: there is no accurate replicase without a large genome and there could be no large genome without an accurate replicase. Thus, the information that can be reliably replicated is less than the information necessary to code for the replicating machinery being composed of strings of the same digit multiplicity. Various ways of resolving this paradox have been proposed and are being worked at. One of the current difficulties in modeling evolutionary population dynamics is to properly outline the scope of “selectability” of replicators, that is, the emergence of Darwinian selection through the extinction of competing sub-populations of coexisting replicators while maintaining the survivors stable, for example, stable against parasites, yet still evolvable over space and time in the sense that the survivors may integrate more different replicating (memory) polymers without making the whole system collapse. It turns out from the research of the past decade that spatially explicit systems of cooperating replicators, that are irrevocably coupled to (“fed by”) metabolic reaction networks, are incomparingly more robust than replicators devoid of metabolism. The intrinsic coupling of translation and replication in reflexive genetic information systems, thus comprising genes whose expression by rules can, in turn, execute those expression rules, are particularly effective and fast in dynamically stabilising the robustness of evolving replicator systems.

One of the most remaining problems is the intrinsic molecular trait of macromolecular low-digit polymers originating from the obligatory mutual affinity between template molecule and product molecule. Macromolecules usually replicate in the parabolic growth regime in which every generation of replicators produces on the average fewer complementary products per template than the previous generation (per template) [27], a general phenomenon termed “strand inhibition”. The very attribute of low-digit polymers, that makes them relatively easy to replicate through template copying, renders them too slow growing in numbers required to open the gates for truly competitive population dynamics, thus for Darwinian selection to apply. What transpires most out of this dilemma is the need for polymers of high catalytic potential, much higher than that of low-digit memory polymers (see Supplement to ref. [9]). Not only are low-digit polymers too inefficient in catalysing the attachment of codon-cognate high-digit monomers to low-digit polymers needed for an operational genetic code (specific aminoacylation of transfer RNA), this requires a high degree of selectivity with respect to the recognition of, both, a high variety of high-digit monomers and low-digit polymers (amino acids and transfer RNA), which can only be accomplished by high-digit polymers (proteic aminoacyl RNA synthetases). In addition and most importantly, only high-digit polymers, by virtue of providing efficient replication machineries (proficiently selective catalysts), can bring down the residual intrinsic error-proneness of low-digit polymers to levels that resolve Eigen’s paradox and heave the units that harbour them (cells) into the exponential growth regime, where different units can compete with one another and thrive through mutation and selection. In an origin of life context, this is the most fundamental reason for translation to occur at an early stage of evolution. In the wording of a biochemist: nucleic acid helicases and polymerases (protein enzymes), that open up nucleic acid double-strands and, respectively, insert highly selectively the complementary mononucleotides to each template-bound primer strand within the same generation—resulting in exponential growth of dynamically stable populations—are needed very early on in an evolutionary timescale. For this to happen, the production of highly diverse gene products is extremely advantageous (Figure 2), if not mandatory [9,10]. 

On the other hand, the extant translation machinery itself is mainly composed of few long low-digit polymers (ribosomal RNA) and an optimal number of uniformly small high-digit polymers (ribosomal proteins). It turns out that the highest possible efficiency of production of the translation machinery, that is, the need to sequester as little autocatalytic enzymatic time as possible to synthesise this machinery, in order to have as much available time as possible to produce other catalysts than itself, is the guiding concept for the fact that ribosomes are mainly composed of RNA with a high rRNA/r-protein ratio [28]. The time ribosomes invest in r-protein synthesis can be up to two orders of magnitude longer than for an equivalent mass of rRNA, especially in fast growing organisms.

Alternative nucleic acids composed of fewer letters, *bits* or *trits* rather than *quits*, could be considered in extra-terrestrial biota and/or during early periods of the origin of life on Earth. They might encode a smaller or larger choice of proteinogenic amino acids—or some other molecular equivalent of a functionally more diverse polymer than nucleic acids—by translating from shorter or, respectively, longer *bytes* or *trytes* as mentioned above and shown in Figure 1A,B. In principle, alternative nucleic acids could also form triple complements through triple-strand formation or even higher-order supramolecular string associations, which would change the stoichiometry of transcription and replication. The chemical reality, as expressed in pairing/tripling/quadrupling/… properties of such alternative nucleic acids, would be expected to impose grave consequences on their copying and translation fidelity, and thus on the number of genes and maximal genome length [23,24,25]. In principle, memory strings could also be extended to higher than quaternary digit multiplicities (not to be confused with a locally “expanded alphabet” of triplet codons) and translated using longer than 4-digit blocks (pentaplet, hexaplet etc. codons). In the reality of macromolecules offered by nature, however, more diverse higher-digit memory polymers are likely to be copied more erroneously, since the monomers would necessarily be more similar to one another, again limiting the replication error threshold, maximal number of genes and total genome length. In addition, longer codons than quadruplets are at higher risk of being misread due to spontaneous frameshifting and mispairing, which would produce more erroneously assembled proteins (secondary polymers) and necessitate a more elaborate and costly error correction effort by the system.

Yet alien biota that would provide linear memory polymers that were markedly more rigid than “natural” RNA or DNA, thus perhaps less prone to frameshifting and mispairing, should not be ruled out a priori, not for chemical reasons. The overall energetic cost at the available energy influx needed to generate such polymers, to keep their replication error threshold high, also to keep the erroneously produced secondary polymers under a liveable limit, are probably much more preventive factors than the huge choice of bricks that chemistry can in principle offer.

## 6. Conclusions

The chemistry on our planet apparently produced prebiotic bricks that could condense under prebiotic reaction conditions into 1D polymers (nucleic acids) that could form double-strands, at least locally in certain string zones, through the spontaneous association (hydrization) of pairwise complementary digits, as shown for the complement-copying in Figure 1C. The digit multiplicity of the first replicating nucleic acids (*bit*, *trit, quit,* etc.) is unknown, although there is a consensus on *bit* polymers having preceded modern natural (biotic) nucleic acids that are generally *quit* polymers. These prebiotic bricks are purine and pyrimidine ribonucleoside pairs that, under appropriate prebiotic reaction conditions being present on this planet some 3.6 Gya, could condense with phosphate and polymerize into RNA and similar RNA-like linear polymers [21]. At least a part of the early nucleic acid single strands could synthesize 3*Q*-translated secondary polymers (Figure 2), viz. polypeptides and proteins very early on, or else we would hardly expect the genetic code to be universal [29]. Apparently, on Earth, RNA proved to be the most successful “primary” memory polymer. Not only can its monomer sequence be easily copied and faithfully reproduced. More faithful and streamlined information storage carriers can be derived from RNA by its deoxygenation to DNA. Most importantly, RNA not DNA can direct and catalyse the linking of amino acids into defined strings of polypeptides, that is, take an essential part in catalysing the controlled dehydration of amino acids to produce amide bonds, a process termed peptidyl transfer (PT). Strong evidence suggests that uncoded PT preceded coded PT, thus, that RNA could grow polypeptide chains from amino acids before a recognition system eventually emerged—from RNA, too—that allowed RNA-directed PT to profit from specific codon–anticodon interactions, and thus to translate genetic information [30,31,32]. 

The arguments presented in this work insinuate that in other prebiotic environments perhaps different kinds of linear polymers could become dominant and evolve in reproducing entities, and this should not be excluded a priori from a chemical-molecular perspective. But we should expect alien and very early biota to evolve right from the start string polymers of both kinds, low-digit and high-digit variants, where the more diverse latter is encoded by the simpler former.

## Figures and Tables

**Figure 1 life-09-00017-f001:**
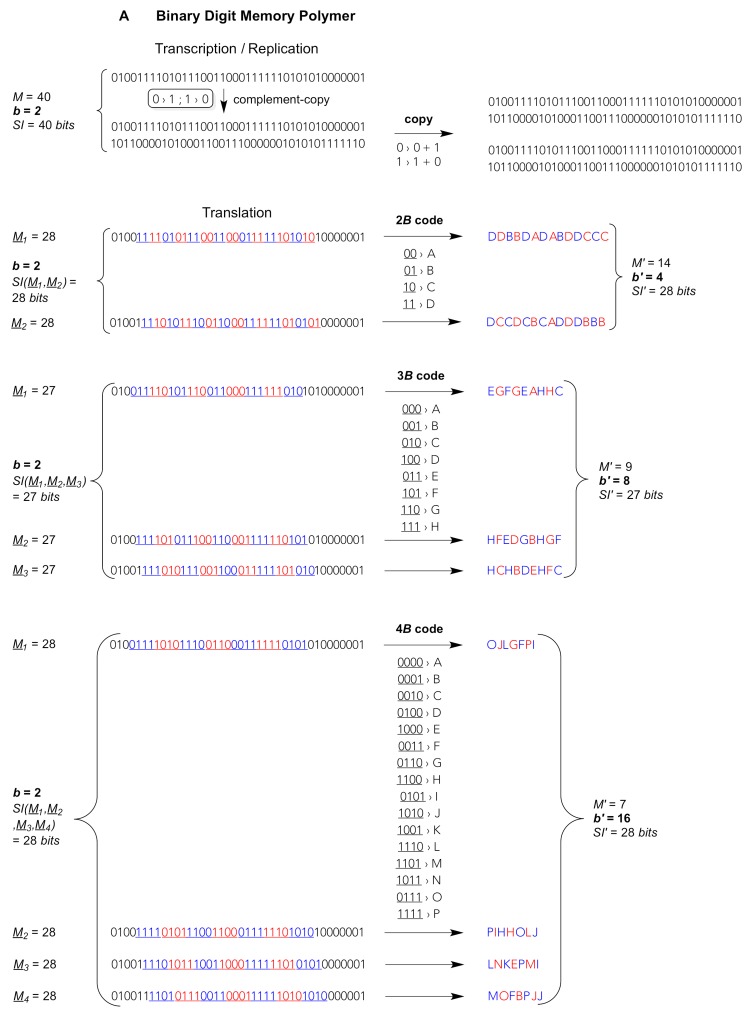
**Template-copying and translating an exemplary string of total *M* = 40 residues** composed of *b* different digits transmitting, according to Equation (3), a given amount of Shannon Information *SI(M)* from one parental string to others. **Upper part**: Transcription/Replication. A single complement-copying event (vertical arrow) is realised during, both, transcription (or reverse transcription) and the first step of replication; the second (horizontal) step applies to replication only. Complement-copying rules (small round-edged shaded frames) by virtue of a minimal requirement for self-complementary digits, i.e., *bits* {0,1}, *trits* {0,1,2} and *quits* {0,1,2,3}. **Lower part**: Translation of consecutive blocks (*B* = *bytes*, *T* = *trytes*, *Q* = *quytes*) of 2–4 digits (**A**
*bits*, **B**
*trits*, **C**
*quits*). The reading frame (underligned coloured digits) is translated into products (strings of letters) of a condensed residue number *M’*, higher digit multiplicity (diversity) *b’* = *b*^2–4^, and unchanged *SI’*. Frameshifts *M*_1–2_ for 2-digit blocks, *M*_1–3_ for 3-digit blocks and *M*_1–4_ for 4-digit blocks generate alternative translation products of the same length *M’*, diversity *b’* and *SI’* but radically different sequences. (**A**) Binary digit (*bit*) strings, replicated and translated from 2–*bit bytes* (2*B*), 3–*bit bytes* (3*B*) and 4–*bit bytes* (4*B*). (**B**) Ternary digit (*trit*) strings, replicated and translated from 2–*trit trytes* (2*T*), 3–*trit trytes* (3*T*) and 4–*trit trytes* (4*T*). (**C**) Quaternary digit (*quit*) strings, replicated and translated from 2–*quit quytes* (2*Q*), 3–*quit quytes* (3*Q*) and 4–*trit quytes* (4*Q,* code and translation products not shown). Shadowed large frame: the current natural (biotic) memory system are *quit* strings being translated from reading frames of consecutive 3*Q* utilizing *b’* = 64 codons that are reduced, mainly for fidelity reasons, to *b’_eff_* = 20 effectively translated digits, viz. the “universal genetic code” for 20 different amino acids and a stop signal (i.e., lack of amino acid). Reproduced and modified from The Handbook of Astrobiology; published by CRC Press, 2019 © Taylor & Francis [21].

**Figure 2 life-09-00017-f002:**
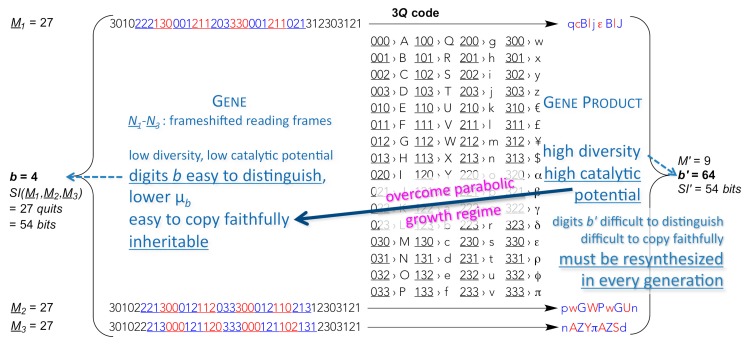
Translation of parts of low-digit memory polymers into high-digit functional polymers as a means to achieve inheritable exponential population growth. The expectedly much higher catalytic potential of high-digit polymers (gene products) allows for more efficient use of the templating ability of the low-digit memory polymers (containing genes), both in terms of copying fidelity *Q* = (1–µ*_b_*)*^M^*, where µ*_b_* denotes mutation probability of every digit, and population growth order *p*, where 0 < *p* < 1 defines the parabolic growth regime and *p* = 1 the exponential growth regime, in populations of replicators *x_i_* that grow in time *t*, as in *dx_i_/dt* = *k_i_ x_i_^p^*, where each replicator population *i* replicates with an apparent replication rate konstant *k_i_.*

**Table 1 life-09-00017-t001:** String lengths *M*, binary string compression factors ^‡^ and approximate *SI* values for strings of different integer-digit multiplicities ^#^ bearing approximately the same storage capacity *N* ≈ 10^12 §^.

Digit Multiplicity ^#^			String Length *M*	String Compression ^‡^ *M_b = 2_/M_b≥2_*	Shannon Information (Equation (3), *b’_eff_*: →*b’*: →*b*)	Storage Capacity *N* ^§^ (Equation (2), *b’_eff_*: →*b’*: →*b*)
*b*	*b’*	*b’_eff_*				
2			40	1.00	40 *bits*	2^40^ ≈ 10^12^
3			26	1.54	≈ 41.2 *bits* = 26 *trits*	3^26^ ≈ 10^12^
4			20	2.00	40 *bits* = 20 *quits*	4^20^ ≈ 10^12^
5			17-18	2.35–2.22	≈ 39.5–41.8 *bits*	5^17–18^ ≈ 10^11–12^
	8		13-14	3.08–2.85	≈ 39.0–42.0 *bits*	8^13–14^ ≈ 10^11–12^
	9		12-13	3.33–3.08	≈ 38.0–41.2 *bits*	9^12–13^ ≈ 10^11–12^
	16		10	4.00	40 *bits*	16^10^ ≈ 10^12^
		20	9-10	4.44–4.00	≈ 38.9–43.2 *bits*	20^9–10^ ≈ 10^11–13^
	27		9	4.44	≈ 42.8 *bits*	27^9^ ≈ 10^12^
	64		6-7	6.67–5.71	≈ 36.0–42.0 *bits*	64^6–7^ ≈ 10^11–12^
	81		6	6.67	≈ 38.0 *bits*	81^6^ ≈ 10^11–12^
	256		5	8.00	40 *bits*	256^5^ ≈ 10^12^

# *b’* = *b*^2^, *b*^3^, *b*^4^ (three different codon lengths); *b’_eff_* = reduced through redundancy from higher *b’*, cf. Section 3. ‡ cf. complexity Ψ = *bits* per monomer (e.g. per nucleotide, codon, amino acid) [9]. § precision within ±1 order of magnitude.

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
