# Peer review of "Low-Digit and High-Digit Polymers in the Origin of Life"

_life, 2019, doi:10.3390/life9010017_

Round 1
Reviewer 1 Report
The manuscript by Strazewski entitled “low Low-Digit and High-Digit Polymers in the Origin of Life” discusses from an information theory background the evolvement of life from low-digit to high-digit polymers with immediate request on a versatile high digit-composed replication machinery for the low digit polymers. This is a very well written interesting paper recommended for publication.
Minor points:
(i) It is suggested to improve the abstract by including the uni-directional nature of information transfer from RNA -> AA as well as the Eigen paradox.
(ii) The citations are focused on the RNA side. There is recent progress on the peptide world theory that may serve for the requested RNA replication machinery.
Author Response
Thank you for your comments.
(i) I added in the abstract the following text:
"The fine-grained diversity of proteins makes an unambiguous information transfer from protein templates too error-prone, so they need to be resynthesized in every generation. But proteins can catalyse, both, their own reproduction as well as the efficient and faithful replication of nucleic acids, which resolves in a most straightforward way an issue termed “Eigen’s paradox”."
(ii) I decided to add an important reference [27] published by S. Reuveni, M. Ehrenberg, J. Paulsson: Ribosomes are optimized for autocatalytic production. (Nature 2017, 547, 293-297), and the following paragraph shortly before Figure 2, after "... For this to happen, the production of highly diverse gene products is extremely advantageous (Figure 2), if not mandatory [9,10].
On the other hand, the extant translation machinery itself is mainly composed of few long low-digit polymers (ribosomal RNA) and an optimal number of uniformly small high-digit polymers (ribosomal proteins). It turns out that the highest possible efficiency of production of the translation machinery, that is, the need to sequester as little autocatalytic enzymatic time as possible to synthesise this machinery, in order to have as much available time as possible to produce other catalysts than itself, is the guiding concept for the fact that ribosomes are mainly composed of RNA with a high rRNA/r-protein ratio [27]. The time ribosomes invest in r-protein synthesis can be up to two orders of magnitude longer than for an equivalent mass of rRNA, especially in fast growing organisms."
I prefer not to write about a "peptide world" theory.
Reviewer 2 Report
This is an interesting and well written paper. I think that it should be accepted for publication in "Life".
Two remarks:
# Is it so obvious that nucleic acids (NA's) contain less "diversity" that proteins? It is true if one consider NA's divided into their 4 possible monomeric bricks (which is chemically true), but not if one consider them as series of the 64 possible codons, which are their true significant - informative - constituents. From the later point of view, NA's are more "diversified" than proteins.
# The author might introduce a short discussion about non-ribosomal peptides, hence about the translation of peptides into peptides. How would he consider this alternative information transfert in his "low-digit - high-digit theory"? Could we speak of a "high digit - high digit" translation ? A non-ribosomal code as been proposed, in which 10 aa's seem to be necessary to choose a single aa residue in the peptide to be synthesized.
Author Response
Thank you for your comments.
Response to remark 1: In my opinion the structural diversity of proteins is evidently higher than that of nucleic acids, especially the density of structural diversity. Recent work by C. W. Carter, Jr., P. R. Wills. Interdependence, reflexivity, fidelity, impedance matching, and the evolution of genetic coding (Mol. Biol. Evol. 2018, 35, 269-286) elaborates on this point further. I cite their work (including their supplemental material) as ref. [9] several times in the context of diversity. To make my point clearer, I added in the abstract:
Nucleic acids are linear polymers composed of a relatively low structural variety of monomeric residues, thus, a low diversity per accessed volume. Proteins are more compact linear polymers that dispose of a huge compositional diversity even at the monomeric level, thus, bear a much higher catalytic potential.
Response to remark 2: I did mentioned that...
"Nature can reliably transmit information, being imprinted into molecular atom arrangements under liquid water-conditions only from low-digit to low-digit, or from low-digit to high-digit polymers, never from high-digit to low-digit polymers. These low-digit read-outs from high-digit polymers would immediately loose their informational identity. Molecular recognition of high-digit polymers, thus from highly diverse molecular variants, is too ambiguous."
I prefer not to dilute the text with discussing non-ribosomal peptides. In my opinion, the coding of amino acids by peptides is not a general phenomenon, can only work idiosyncratically (if at all), for the reasons given above (error-proneness).